# Self-Supervised Graph Attention Collaborative Filtering for Recommendation

**Jiangqiang Zhu** [1], **Kai Li** [1,2,*], **Jinjia Peng** [1,2] and **Jing Qi** [1,2]

1    School of Cyber Security and Computer, Hebei University, Baoding 071000, China
2    Hebei Machine Vision Engineering Research Center, Hebei University, Baoding 071000, China
*    Correspondence: likai@hbu.edu.cn

**Abstract:** Due to the complementary nature of graph neural networks and structured data in recommendations, recommendation systems using graph neural network techniques have become mainstream. However, there are still problems, such as sparse supervised signals and interaction noise, in the recommendation task. Therefore, this paper proposes a self-supervised graph attention collaborative filtering for recommendation (SGACF). The correlation between adjacent nodes is deeply mined using a multi-head graph attention network to obtain accurate node representations. It is worth noting that self-supervised learning is brought in as an auxiliary task in the recommendation, where the supervision task is the main task. It assists model training for supervised tasks. A multi-view of a node is generated by the graph data-augmentation method. We maximize the consistency between its different views compared to the views of the same node and minimize the consistency between its different views compared to the views of other nodes. In this paper, the effectiveness of the method is illustrated by abundant experiments on three public datasets. The results show its significant improvement in the accuracy of the long-tail item recommendation and the robustness of the model.

**Keywords:** recommendation system; collaborative filtering; graph neural networks; self-supervised learning; multi-task learning

## 1. Introduction

The internet's fast expansion and the admission of different sectors into the era of digital economy have produced a tremendous amount of data. However, not all of these data are valuable, and it becomes very difficult for users to obtain useful messages out of a huge volume of data. To address the issue of information overload, recommendation systems supply users with individualized requirements by mining effective information on their behavioral data. Early recommendation algorithms mainly include collaborative filtering-based recommendation [1,2], logistic regression-based recommendation, factorization machines [3,4] based recommendation and recommendation combined with gradient boosting decision tree [5]. Collaborative filtering is a popular paradigm and includes both item collaborative filtering and user collaborative filtering algorithms. In order to be able to better handle sparse data and improve the model's generalization, the recommendation based on matrix factorization [6] is derived from collaborative filtering. Compared to collaborative filtering-based recommendations that utilize only the displayed or implicit feedback information between users and items, logistic regression-based recommendations are able to utilize and incorporate more user, item, and contextual features. Factorization machine-based recommendation augments classical logistic regression with a second-order component, giving the model the ability to incorporate features. To combine the advantages of multiple models, Facebook [5] combines logistic regression and gradient boosting decision tree to propose a combined GBDT+LR model.

Deep learning-based recommendation models have received a lot of attention as the technology has advanced. Compared with traditional machine learning models, deep

learning models are more expressive and can mine more latent patterns in data. The model structure based on deep learning is very flexible. It may be dynamically altered based on the business scenario and data characteristics to ensure that the model fits precisely with the application scenario. Recommendation based on deep learning has become mainstream, from the simple single-layer neural network model AutoRec [7] to the classical deep neural network structure Deep Crossing [8], which mainly increases deep neural network layer count and structural complexity. NeuralCF [9] alters the interaction of user and item vectors and enriches the way features are intersected in deep learning. Weed&Deep [10] enhances the model's integrative capabilities by integrating two deep learning networks with distinct traits and complementing strengths. NFM [11] utilizes neural networks to improve the feature crossover capability of the second-order part.

In recent years, graph neural networks [12–15] have received a lot of interest from academia and business due to the powerful representational power of graph structures. Graph neural networks are a class of deep learning-based methods for processing graph domain information, and recommendations combined with graph neural networks have been widely studied due to their better representational power and interpretability. Collaborative filtering recommendation based on graph neural networks [16,17] builds the user–item interaction as a user–item bipartite graph. It utilizes higher-order connectivity on the bipartite graph to enrich the user and item vector representation. Recommender with graph convolutional networks [16] provides a complete solution for including higher-order neighbors in node representation learning. Although effective, there are still some limitations: sparse supervised signals, long-tail problem and interaction noise. Most models are performed in a supervised learning paradigm for recommendation tasks [9,18], where supervised signals are derived from observed user–item interactions. However, the observed interaction information is extremely sparse compared to the entire interaction space [19,20] and is not sufficient to learn feature-rich node representations. The long-tail problem has resulted in a high degree of nodes (abundant number of connected edges) dominating representation learning [21], and low degrees of nodes (scarcity of connected edges, i.e., long-tailed items) are difficult to learn. Most of the feedback provided by users is implicit (e.g., click and browse) as opposed to explicit (e.g., rate, purchase, and like). Thus, observed interactions usually contain noisy data [22]. For example, users unintentionally click or browse content that does not interest them, and aggregation methods in graph convolutional networks are unable to distinguish these noisy data, making the learning of node representations more susceptible to noisy data.

This paper addresses the above limitations by combining graph attention networks [23] and self-supervised learning [24]. As a backbone network for supervised learning tasks, the graph attention network is implemented. It can be implemented to assign different learning weights to different neighboring nodes, which significantly decreases the problem of bringing in noisy data to the aggregation process. Self-supervised learning is widely utilized in the domains of computer vision and natural language processing [25,26], but is currently relatively rare in the field of recommendation. At its core is a framework called proxy tasks, which allows the utilization of unlabeled data itself to generate labels without the need for manual data annotation. For example, Bert [27] masked some of the words in the text utilizing a random mask and set a proxy task to predict them; RotNet [28] utilizes the rotated image as the input to the training model, giving the model better representation capabilities. In contrast to supervised learning, self-supervised learning permits changes in the input data to leverage the unlabeled data space to achieve significant improvements in downstream tasks. In this study, the benefits of self-supervised learning are included in the recommendation representation learning to solve the constraints of the graph neural network-based recommendation models mentioned above.

The self-supervised learning task contains two main parts: (1) data augmentation to generate multi-views per node, and (2) contrastive learning is used to maximize consistency between multiple views of the same node while minimizing consistency across views of distinct nodes. In graph representation learning, the properties of the data have a strong

impact on the representation results of the nodes, especially their structural properties. Therefore, the data without labels can be built by altering the structure of the graph. To this end, this paper utilizes three graph data-augmentation methods of node mask, edge mask, and layer mask to change the graph structure and perform contrastive learning based on graph attention networks. Self-supervised learning enhances node representation learning by investigating the interactions within nodes. Thereby, self-supervised learning complements graph neural network-based recommendation models. Node self-identification provides auxiliary supervised signals that complement classical supervised learning from observed interactions only. Graph data augmentation reduces the impact on model training by reducing the edges of high-degree nodes.

In summary, this work proposes self-supervised graph attention collaborative filtering for recommendation. It can effectively solve the problems of sparse supervised signals and long tails in graph neural network-based recommendations and reduce the impact from the drought-in interaction noise data. The following details the proposed method and demonstrates its effectiveness through extensive experiments. Section 3 states the proposed method in detail and contains two main tasks: supervised task and self-supervised task. Sections 4 and 5 demonstrate the effectiveness of the proposed method through extensive experiments on three public datasets.

## 2. Related Work

This subsection introduces three aspects related to this work: collaborative filtering-based recommendation, graph neural network-based recommendation, and self-supervised learning.

### 2.1. Collaborative Filtering-Based Recommendation

Collaborative filtering-based recommendation systems implement recommendation tasks by calculating the similarity between users or items, which assume that users who have interacted with the same item have similar interests. They can be generally classified into memory-based collaborative filtering recommendations [2] and model-based collaborative filtering recommendations [29–31]. Memory-based collaborative filtering recommendation usually utilizes the nearest neighbor idea to calculate similarity using the historical interaction information of users or items. Common methods for similarity include the Pearson correlation coefficient [32], cosine similarity [33], Jaccard similarity coefficient, and Euclidean distance. The model-based collaborative filtering recommendation simulates users' ratings of items by modeling, which uses machine learning or deep learning techniques to construct models and employs a large amount of data trained for the recommendation task.

### 2.2. Graph Neural Network-Based Recommendation

Although deep learning-based recommendation systems have achieved positive results, the prediction and training paradigms of these methods ignore the higher-order structural information in the data. Therefore, there are still significant limitations. The growth of graph neural networks in the past few years has presented a good idea for overcoming the obstacles in recommendation systems. The graph neural network is based on a user–item bipartite graph by an aggregation function enriched with node embedding representations. By iterative propagation, each node can access higher-order neighbor information instead of only first-order neighbor information as in previous methods. Graph neural network-based approaches have become state of the art in recommender systems due to their advantages in processing structured data and mining structural information. SpectralCF [34] utilizes collaborative filtering with spectral graph convolution; GC-MC [35] and NGCF [16] model graph convolutional networks on the original data space where users and items interact, which is more effective in practical applications; NIA- GCN [36] adds neighbor node interaction awareness to graph convolutional networks; DGCF [17] decouples the user's complex interaction intent, obtains fine-grained embedding represen-

tation and tunes up the interpretability of the model; BGCF [37] treats the interaction graph also as a random variable to mitigate the impact caused by the uncertainty of the user–item interaction graph; and LR-GCCF [38] and LightGCN [39] analyze operations, such as feature transformation and nonlinear activation, in graph neural networks, simplifying them to improve the performance of the model.

Although the above methods achieve relatively good results, the influence of noisy data in the process of aggregating neighbors does not allow obtaining high-quality node representations. Therefore, this work employs a multi-head graph attention network to mine the correlation between neighboring nodes and obtain accurate node representations. This work uses a supervised learning paradigm for model training, but sparse supervised signals lead to a loss of performance. Therefore, self-supervised learning is incorporated as an auxiliary task to enhance the supervised learning task.

*2.3. Self-Supervised Learning*

Self-supervised learning [25,27,40] mainly includes generative and contrastive self-supervised learning. The goal of generative self-supervised learning is to learn a low-dimensional vector representation for the input that can retain as much information as possible. Contrastive self-supervised learning learns a comparative noise contrast estimation [41,42] (NCE) aim, which might be global versus local, or global versus global. The former focuses on constructing relationships between a sample's local and global contextual representations, whereas the latter compares explicitly the multi-views of different samples. Self-supervised learning has been the subject of much related work in the fields of computer vision [25] and natural language processing [27]. In contrast to image and text data, self-supervised learning also helps to understand structural and attribute information in graphical data. Thus, self-supervised learning is also applicable to graph-structured data. For example, InfoGraph [43] and DGI [44] learn node representations on mutual information among nodes and local structures; Hu [45] and others extend this approach by training a graph convolution-based model to learn node representations; Kaveh [46] learns node and graph representations via a contrastive paradigm, comparing representations of one view with those of another; GCC [47] utilizes subgraph discrimination as a pre-training task and then improves the representation capability of the graph neural network via contrast learning.

There is not much recommendation-related work using self-supervised learning. DSSR [48] performs self-supervised learning in the latent space to promote convergence for sequential recommendations; Google [24] utilizes a multi-task framework, employing a deep neural network with dual towers as encoders. This paper also utilizes a multi-task framework, which differs from the graph-based recommendation and uses only ID as a feature. HCCF [49] jointly captures local and global collaborative relationships through a hypergraph-enhanced cross-view contrastive learning framework.

## 3. Proposed Method

This paper proposes the self-supervised learning graph attention-based collaborative filtering recommendation (SGACF), whose architecture is shown in Figure 1. The framework is divided into two components: supervised tasks and self-supervised tasks. The supervised task serves as the main part of the framework, with the graph attention network as the backbone network. The self-supervised task mainly constructs supervised signals from the correlations within the input data and performs joint learning with the supervised task as an auxiliary task. This chapter introduces the supervised task framework and self-supervised tasks. It describes how data augmentation in self-supervised learning is performed to generate multiple view representations, and then contrastive learning is performed to construct the pretext task based on the generated representations. Finally, a theoretical analysis of how self-supervised tasks enhance supervised learning is presented.

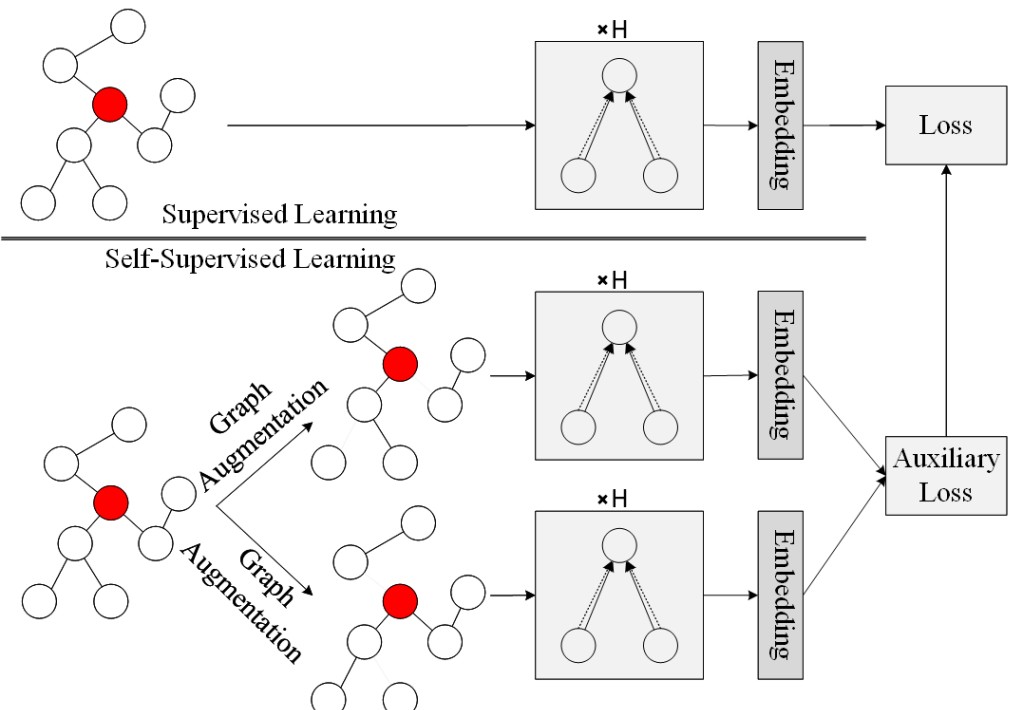

**Figure 1.** Overall framework. The upper part is for the supervised task, and an H-graph attention network does feature extraction with multiple lines in the network representing a multi-headed attention mechanism; the lower part is self-supervised learning as an auxiliary task, sharing parameters with the network layer in the supervised task.

### 3.1. Supervised Learning

The primary flow of the supervised task is described in this subsection, as shown in Figure 2. It is made up of three major parts: (1) the embedding layer provides initialized vector representations of users and items; (2) the neighbor aggregation and embedding propagation layer generates multiple refined embedding representations of nodes by aggregating higher-order neighbor features and finally synthesizes the final vector representation of nodes; and (3) the prediction layer models user–item interaction and generate the user's preference ratings for objects. Finally, the supervised loss is described.

#### 3.1.1. Embedding Layer

First are the symbols that appear in the pre-defined text. The sets of users and items are described by $\mathcal{U}$ and $\mathcal{I}$, respectively. $\mathcal{O}^+ = \{y_{ui} | u \in \mathcal{U}, i \in \mathcal{I}\}$ is considered as an observed interaction, and $y_{ui}$ indicates that user $u$ has previously interacted with item $i$. This paper builds the interaction between users and items as a user–item bipartite graph $\mathcal{G} = (\mathcal{V}, \mathcal{E})$, with the set of nodes $\mathcal{V} = \mathcal{U} \cup \mathcal{I}$ containing all users and items and the set of edges $\mathcal{E} = \mathcal{O}^+$ representing the observed interactions. SGACF only utilizes user and item IDs as features and maps them to low-dimensional embedding vectors through the embedding layer. Therefore, in this paper, the set of users and items can be defined as the set of user embedding vectors $E_u$ and the set of item embedding vectors $E_i$ after passing them through the embedding layer as follows:

$$E_u = \{e_u | u \in \mathcal{U}, e_u \in \mathbb{R}^d\}; E_i = \{e_i | i \in \mathcal{I}, e_i \in \mathbb{R}^d\} \tag{1}$$

where $d$ is the size of the embedding vector. Compared to traditional recommendation models that directly employ user and item ID embeddings for interaction modeling, this paper utilizes the characteristics of graph structure to improve the embedding by embedding propagation. This is able to retain the user's own interest and also to explore the potential interest of the user.

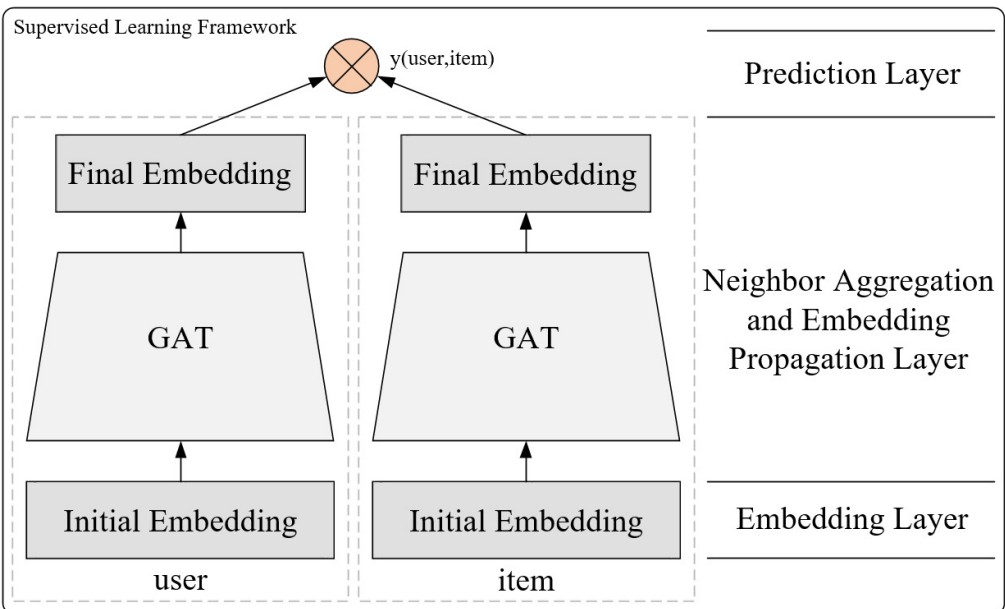

**Figure 2.** Supervised learning framework. It consists of three main components: the embedding layer, the neighbor aggregation and embedding propagation layer, and the prediction layer. y(user,item) is the user's preference score for the item .

### 3.1.2. Feature Propagation Layer

Users and items have generated interactions between them that can serve as feature representations of each other. Therefore, the aggregation operation on the graph structure is crucial to the node vector representation. Previous aggregation operations based on graph convolutional networks neglected to distinguish the degree of influence between adjacent nodes, resulting in a vector representation of nodes that does not satisfy the needs of personalized recommendations. In this paper, graph attention networks are utilized to accomplish the aggregation operation by calculating the self-attention coefficients between neighboring nodes, which are then utilized for the linear combination of features corresponding to them. The simplified formula is expressed as $e'_{u/i} = \mathcal{AGG}(\cdot)$, where $e'_{u/i}$ denotes the new vector representation generated, and $\mathcal{AGG}(\cdot)$ is the aggregation function. The aggregation function is implemented below. First, to obtain sufficient representational power, the input node embedding vectors are feature transformed to obtain a new set of node embedding vectors. A shared parameterized weight matrix $W \in \mathbb{R}^{d' \times d}$ is needed to act on each node, where $d'$ is the size of the transformed vector representation. Then, the attention coefficient is calculated between the neighboring nodes. The formula is as follows:

$$\partial_{ui} = a(We_u, We_i) \tag{2}$$

where $\partial_{ui}$ denotes the importance of item $i$ to user $u$. $a$ denotes a shared attention mechanism. Item $i$ is a first-order neighbor of user $u$. To make the attention coefficients easily comparable across nodes, this paper utilizes the *softmax* function to normalize the importance of all $i$:

$$\alpha_{ui} = softmax_i(\partial_{ui}) = \frac{exp(\partial_{ui})}{\sum_{k \in N_u} exp(\partial_{uk})} \tag{3}$$

where $N_u$ is the set of items that user $u$ has interacted with. In the experiments, $a$ is a feedforward neural network using a weight vector $\overrightarrow{w} \in \mathbb{R}^{2d'}$ parameterized by a single layer and using the *LeakyReLU* nonlinear activation function (with negative input slope is 0.2). The complete formula for the attention coefficient is as follows:

$$\alpha_{ui} = \frac{exp(LeakyReLU(\overrightarrow{w}^T[We_u || We_i]))}{\sum_{k \in N_u} exp(LeakyReLU(\overrightarrow{w}^T[We_u || We_k]))} \tag{4}$$

where $\cdot^T$ denotes transposition and $||$ indicates a concatenation operation. Multi-headed self-attention is required to steady the procedure of self-attention learning. Specifically, the features output by $K$ independent self-attentive mechanisms are averaged, and a nonlinear operation is added. The specific formula is as follows.

$$e_u^{final} = \sigma\left(\frac{1}{K}\sum_{k=1}^{K}\sum_{j\in N_u}\alpha_{uj}^k w^k e_j\right) \tag{5}$$

where $||$ denotes the concatenation operation, $\alpha_{uj}^k$ denotes the coefficient of the $k$th attention mechanism output between user $u$ and item $i$, and $w^k$ is the weight matrix of the corresponding linear transformation.

The node representation is enhanced by a first-order neighbor propagation layer, and then multiple graph attention network layers are used to obtain higher-order neighbor features. Such higher-order neighbor features can dig into the potential interests of users and can effectively improve the generalization of the model. Each node in the interaction graph performs first-order propagation to update the node representation, and the second-order neighbor features can be obtained by iteratively performing first-order propagation. Thus, the synergistic signals of higher-order neighbors can be obtained through multiple iterations. The specific formula is expressed as follows:

$$\begin{cases} e_u^{(1)} = \mathcal{AGG}(e_u^{(0)}, \mathcal{G}) \\ e_u^{(h)} = \mathcal{AGG}(e_u^{(h-1)}, \mathcal{G}) \end{cases} \tag{6}$$

where $e_u^{(0)}$ is the node vector representation after initialization, $e_u^{(1)}$ is the node representation after aggregating first-order neighbors, and $e_u^{(h)}$ is the vector representation after aggregating h-order. The output vectors of multiple networks contain node vector representations with different order neighbor features, which directly affect the final vector representation of the nodes. Nodes with rich low-order neighbors are less dependent on higher-order neighbor collaboration signals, and conversely, require more high-order neighbor signals to enrich the vector representation. Therefore, averaging pooling is adopted to merge the vector representations of nodes of different orders. The formula is as follows:

$$e_u^{final} = \frac{1}{H+1}\sum_{h=0}^{H} e_u^h \tag{7}$$

### 3.1.3. Prediction Layer

SGACF utilizes graph attention networks to obtain node vector representations of users and items, and then embeds higher-order neighbor co-signals into its representation according to the higher-order connectivity principle, and finally models the interaction between users and items by inner product. Then, the preference of user $u$ for item $i$ is

$$\hat{y}_{ui} = (e_u^{final})^T e_i^{final} \tag{8}$$

where $e_u^{final}$ and $e_i^{final}$ are the final vector representations of users and items, $\hat{y}_{ui}$ is the preference score of user $u$ for item $i$.

### 3.1.4. Loss of Supervision Task

Pairwise Bayesian personalized ranking (BPR) [18] loss is extensively utilized for recommendation tasks. Its assumption is that the preference behaviors of each user are independent of each other, and the preferences of the same user for different items are independent of each other. BPR takes into account the relative order of observed and

unobserved user–items, and it assumes that observed interactions are more indicative of user preferences and therefore are granted a higher ranking than unobserved interactions:

$$\mathcal{L}_{BPR} = \sum_{(u,i^+,i^-)\in\mathcal{O}} -ln\sigma(y_{ui^+} - y_{ui^-}) \tag{9}$$

where $\mathcal{O} = \{(u,i^+,i^-)|(u,i^+)\in\mathcal{O}^+,(u,i^-)\in\mathcal{O}^-\}$ denotes paired training data. $i^+$ and $i^-$ denote positive and negative samples, respectively. $\mathcal{O}^+$ denotes unobserved samples. $\sigma$ is the sigmoid activation function, and *Adam* [50] is applied to optimize the model to update the model parameters.

$$\hat{y}_{ui} = (e_u^{final})^T e_i^{final} \tag{10}$$

### 3.2. Data Augmentation of Graph Structures

Data augmentation is a technique that allows limited data to produce more equivalent data to extend the dataset. The main data-augmentation techniques commonly utilized in the field of computer vision are geometric transformation, color adjustment, style transfer, adding noise, etc., which can generate a huge amount of equivalent photos. As a result of the non-Euclidean nature of the graph structure, it is hard to directly adapt data-augmentation methodologies from the vision domain to the graph data domain. The user–item bipartite graph is built on the basis of user–item interaction and contains collaborative signals. Therefore, this paper utilizes a graph structure-based data-augmentation method, including node mask (NM), edge mask (EM) and layer mask (LM) to create different node views. The data-augmentation method is shown in Figure 3. The method of augmenting graph data may be described symbolically as follows:

$$e_1^{(h)} = \mathcal{AGG}(e_1^{(h-1)}, s_1(\mathcal{G})), e_2^{(h)} = \mathcal{AGG}(e_2^{(h-1)}, s_2(\mathcal{G})),\ s_1, s_2 \sim A \tag{11}$$

where the operations $s_1$ and $s_2$ are executed on $\mathcal{G}$ to change the graph structure and create two related views of node $e_1^{(h)}$ and node $e_2^{(h)}$. Setting the probability of a node being dropped as $\rho$, $s_1$ and $s_2$ can be modeled as follows.

#### 3.2.1. Node Mask (NM)

The probability of nodes and connected edges mask is $\rho$. $s_1$ and $s_2$ are modeled as follows:

$$s_1(\mathcal{G}) = (C' \odot \mathcal{V}, \mathcal{E}), s_2(\mathcal{G}) = (C'' \odot \mathcal{V}, \mathcal{E}) \tag{12}$$

where $C', C'' \in \{0,1\}^{|\mathcal{V}|}$ are two mask vectors applied to node set $\mathcal{V}$. It gathers some nodes and their edges from random shaded nodes to generate two subgraphs. This method may observe the dominant nodes in the graph structure, which improves the robustness of the representation learning to structure.

#### 3.2.2. Edge Mask (EM)

Setting the probability of an edge mask as $\rho$, $s_1$ and $s_2$ can be modeled as

$$s_1(\mathcal{G}) = (\mathcal{V}, C_1 \odot \mathcal{E}), s_2(\mathcal{G}) = (\mathcal{V}, C_2 \odot \mathcal{E}) \tag{13}$$

where $C_1, C_2 \in \{0,1\}^{|\mathcal{E}|}$ are two mask vectors whose randomly masked edge set $\mathcal{E}$ generates two subgraphs. Local neighbors of nodes affect representation learning, further mitigating the sensitivity of representation learning to structure.

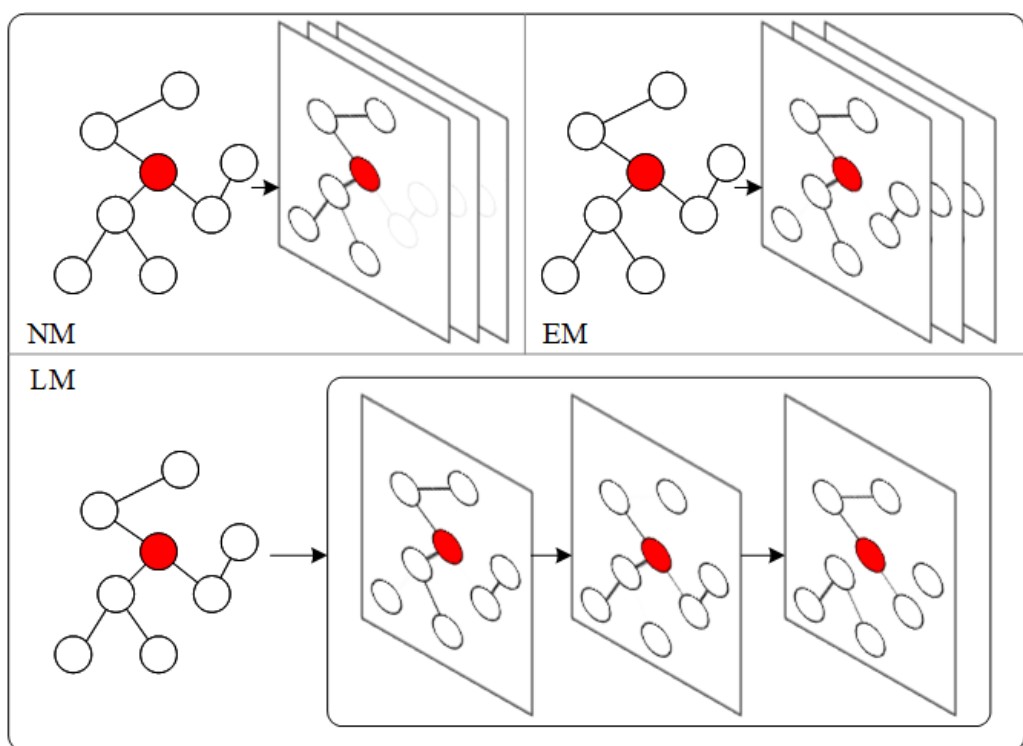

**Figure 3.** Graph augmentation method. Top left is node mask, top right is edge mask, and bottom is layer mask.

### 3.2.3. Layer Mask (LM)

The subgraphs generated by node mask and edge mask are shared across network layers. To further improve the robustness of the model to the graph structure, random walk performs edge mask for the graph structure of each layer of the network input data. The mask probability is set for each layer with edge mask and the expression is as follows:

$$s_1(\mathcal{G}) = (\mathcal{V}, C_1^{(h)} \odot \mathcal{E}), s_2(\mathcal{G}) = (\mathcal{V}, C_2^{(h)} \odot \mathcal{E}) \tag{14}$$

where $C_1^{(h)}, C_2^{(h)} \in \{0,1\}^{|\mathcal{E}|}$ are two mask vectors whose randomly masked edge of the h-layer network.

### 3.3. Self-Supervised Contrastive Learning

#### 3.3.1. Contrastive Learning

Contrastive learning is an implementation of self-supervised learning, mainly designed to solve the problem of little or no labeling. Its classical paradigm is a combination of an agent task and an objective function. This paper utilizes contrastive self-supervised learning, which performs contrastive learning based on the views generated by the data-augmentation methods described above. It utilizes information about commonalities and differences between data pairs as supervised signals to assist in the learning of the supervised task.

#### 3.3.2. Loss of Self-Supervision Task

The main goal of contrastive learning is to increase the consistency of two jointly sampled positive sample pairs while minimizing the consistency of two dependently sampled negative sample pairs. Positive sample pairs $(\{(e'_u, e''_u)|u \in \mathcal{U}\})$ are two distinct views of the same node after augmentation, and negative sample pairs $(\{(e'_u, e''_o)|u, o \in \mathcal{U}\})$

are distinct views of distinct nodes after augmentation. This paper follows SimCLR and utilizes the contrastive loss InfoNCE:

$$\mathcal{L}_{ssl}^{user} = \sum_{u \in \mathcal{U}} -log \frac{exp(s(e'_u, e''_u)/\tau)}{\sum_{o \in \mathcal{U}} exp(s(e'_u, e''_o)/\tau)} \tag{15}$$

where $s(\cdot)$ is is the cosine similarity function. $\tau$ is the temperature coefficient. Similarly, the item-side contrastive loss $\mathcal{L}_{ssl}^{item}$ can be obtained. Combining these two losses results in a self-supervised loss $\mathcal{L} = \mathcal{L}_{ssl}^{item} + \mathcal{L}_{ssl}^{user}$.

### 3.4. Joint Learning

The similarity between users and items is the top priority in the recommendation domain. Contrastive learning is concerned with the similarity between user nodes and their variant views, and item nodes and their variant views, and only plays a auxiliary role in the training phase. Therefore, self-supervised learning is optimized as an auxiliary task jointly with the supervised task to form multi-task learning:

$$\mathcal{L} = \mathcal{L}_{supervised} + \lambda_1 \mathcal{L}_{self-supervised} + \lambda_2 ||\theta||_2^2 \tag{16}$$

where $\theta$ is the parameter of supervised learning, and $\lambda_1$ and $\lambda_2$ are the contribution values of hyperparameter control self-supervised loss and $L_2$ regularization.

## 4. Experiment

The work conducts experiments on three publicly available datasets to evaluate the effectiveness of the proposed method. The settings of the parameters in the model are presented, and the performance is analyzed in comparison with other models.

### 4.1. Datasets and Metrics

This work is experimented on three datasets of Yelp2018, Gowalla and Amazon. Yelp2018 is collected from the 2018 Yelp challenge and considers restaurants, shopping centers and hotels as items. Gowalla is a check-in dataset, where each check-in of a user serves as one piece of data in the dataset. Amazon-review is a widely utilized product recommendation dataset, and the work selects amazon-book from the collection. To ensure the quality of the dataset, a certain number of items are selected in the experiment, and Yelp2018 keeps user data with no fewer than 25 interactions, and the other datasets keep no fewer than 20 interactions.

Model performance evaluation for the top-k recommendation task typically utilizes *Recall@k* and normalized discounted cumulative gain (*NDCG@k*), where $k = 20$. Recall calculates the proportion of items in the recommendation list that have interacted with the user to the number of positive samples in the test set. The higher the value of recall, the better the model recommendation performance. The formula is presented below:

$$Recall = \frac{1}{|\mathcal{U}|} \sum_{u \in \mathcal{U}} \frac{|M_u \cap M_u^{test}|}{M_u^{test}} \tag{17}$$

where $M_u$ is the recommendation list and $M_u^{test}$ is the positive sample of user $u$ in the test set. *NDCG* considers the factor of item location in the recommendation list, and the higher the value, the better the recommendation effect. The formula is as follows:

$$NDCG = \frac{1}{|\mathcal{U}|} \sum_{u \in \mathcal{U}} \frac{\sum_{p=1}^{K} \frac{rel(p)}{log(p+1)}}{\sum_{p=1}^{TP} \frac{1}{log(p+1)}} \tag{18}$$

where $rel(\cdot)$ denotes the item correlation calculation, and $TP$ denotes the items in the recommendation list in order of correlation from largest to smallest.

### 4.2. Parameter Settings

The experiment was implemented via PyTorch. The work was initialized using Xavier [51], setting the size of the embedding vector to 64. The learning rate of the model was set to 0.001, and the $L_2$ regularization coefficient was set to $1 \times 10^{-4}$. The self-supervised learning part parameters $\lambda_1$, $\tau$ and $\rho$ were fine-tuned in the following value ranges $\{0.005, 0.01, 0.05, 0.1, 0.5, 1\}$, $\{0.1, 0.2, 0.5, 1.0\}$ and $\{0, 0.1, ...0.5\}$, respectively. The parameters in the comparison model were set by the original paper.

### 4.3. Experiment Analysis

#### 4.3.1. Explore SGACF's Performance on Datasets

This subsection of the paper explores the performance of SGACF and analyzes the impact of graph augmentation and different orders on the model performance. Here, this paper utilizes augmentation strategies based on graph structure: NM, EM, and LM. It explores the performance of models with different orders from 1 to 4. The experimental results are shown in Table 1.

**Table 1.** Explore SGACF performance on three datasets.

| Layer | Method | Gowalla | | Yelp2018 | | Amazon | |
|---|---|---|---|---|---|---|---|
| | | Recall | Ndcg | Recall | Ndcg | Recall | Ndcg |
| 1 | SGACF-NM | **0.2714** | **0.2199** | **0.2150** | **0.1394** | 0.1633 | 0.0896 |
| | SGACF-EM | 0.2711 | 0.2197 | 0.2144 | 0.1392 | **0.1674** | **0.0905** |
| | SGACF-LM | 0.2711 | 0.2197 | 0.2144 | 0.1392 | **0.1674** | **0.0905** |
| 2 | SGACF-NM | 0.2733 | 0.2201 | 0.2112 | 0.1356 | 0.1658 | 0.0904 |
| | SGACF-EM | **0.2749** | **0.2417** | **0.2163** | **0.1400** | **0.1696** | **0.0922** |
| | SGACF-LM | 0.2728 | 0.2241 | 0.2152 | 0.1394 | 0.1690 | 0.0915 |
| 3 | SGACF-NM | 0.2720 | 0.2205 | 0.2146 | 0.1384 | 0.1688 | 0.0908 |
| | SGACF-EM | **0.2753** | **0.2423** | **0.2171** | **0.1402** | **0.1698** | **0.0925** |
| | SGACF-LM | 0.2741 | 0.2412 | 0.2155 | 0.1397 | 0.1696 | 0.0921 |
| 4 | SGACF-NM | 0.2735 | 0.2198 | 0.2107 | 0.1346 | 0.1660 | 0.0905 |
| | SGACF-EM | **0.2750** | **0.2419** | **0.2166** | **0.1400** | **0.1695** | **0.0923** |
| | SGACF-LM | 0.2730 | 0.2242 | 0.2148 | 0.1387 | 0.1692 | 0.0917 |

The bold indicates the best result.

Analysis of the results shows that the edge mask improves the model more significantly. SGACF-EM outperforms SGACF-LM, and SGACF-LM outperforms SGACF-NM, which may be the inherent ability of the edge mask to capture the graph structure. NM may be thought of as a subset of EM in which certain node edges are masked. The performance of SGACF-NM in the experiment is relatively unstable, from which it can be concluded that mask high degree nodes lead to training instability. Analyzing from the aspect of order, a too-low order leads to a lack of information in the vector representation learned by the model, and too-high order leads to a convergence of node representations, which cannot distinguish the vector representation of different nodes. From Table 1, it is clear that taking the 3-order acquires better performance. In addition, self-supervised learning can enhance the generalization ability of the model. That is, contrastive learning between different nodes can alleviate the problem of the over-smoothing of node representations. To obtain a clearer picture of the effect of the reaction order and graph augmentation method on the performance, the results are visualized as shown in Figure 4.

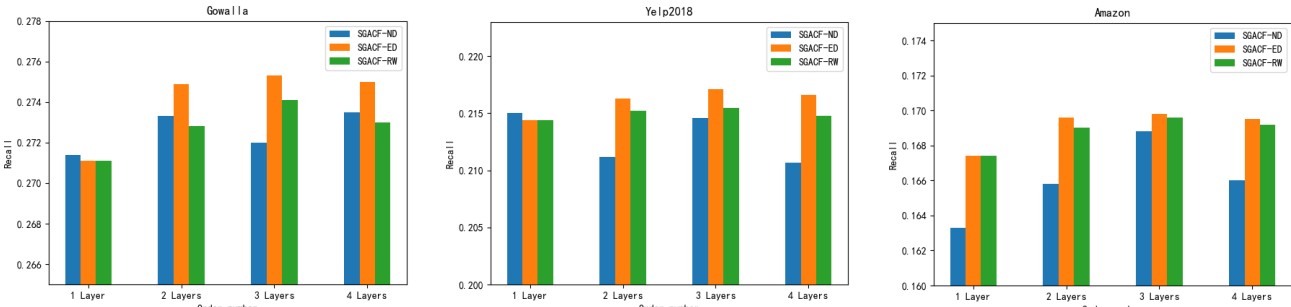

**Figure 4.** Impact of order and graph augmentation on performance.

### 4.3.2. Compared Method

To demonstrate the effectiveness of the model, the work compares the proposed method with the following methods.

- NeuMF [9] combines traditional matrix decomposition and multi-layer perceptron to extract both low- and high-dimensional features. The user's preference score for the item is obtained through the neural network instead of the inner product operation.
- CMN [52] proposes a unified hybrid model which capitalizes on the recent advances in memory networks and neural attention mechanisms for collaborative filtering with implicit feedback.
- SpectralCF [34] proposes a spectral-based convolution operation to construct a deep recommendation model based on spectral collaborative filtering. The model can explicitly mine the higher-order neighborhood features hidden in the interaction graph.
- NGCF [16] utilizes multiple spatial domain graph-based convolutional networks to mine the user's potential interest to obtain multiple representations of the nodes, and then utilizes a weighted summation aggregation function to compute the final vector representation of the nodes.

### 4.3.3. Analysis of Results

This work was experimented with other comparison models on three datasets, and the results are given in Table 2. The performance differences between the comparison models and the model proposed in this paper are analyzed below.

**Table 2.** Performance comparison of SGACF with other comparative models.

| Method | Gowalla | | Yelp2018 | | Amazon | |
|---|---|---|---|---|---|---|
| | **Recall** | **Ndcg** | **Recall** | **Ndcg** | **Recall** | **Ndcg** |
| NeuMF | 0.1402 | 0.1169 | 0.1193 | 0.1040 | 0.1195 | 0.0552 |
| CMN | 0.1746 | 0.1571 | 0.2010 | 0.1075 | 0.1606 | 0.0657 |
| SpectralCF | 0.1756 | 0.1672 | 0.1565 | 0.1209 | 0.0950 | 0.0547 |
| NGCF | 0.2673 | 0.2317 | 0.1993 | 0.1326 | 0.1499 | 0.0866 |
| SGACF-EM | **0.2753** | **0.2423** | **0.2171** | **0.1402** | **0.1698** | **0.0925** |

The bold indicates the best result.

- NeuMF learns only low-dimensional vector representations of users and items utilizing the embedding layer. The results are poor on all three datasets, indicating that there is a considerable paucity of valuable information in the embedding vectors of users and items.
- CMN outperforms other comparison models in terms of recall on the Yelp and Amazon datasets. This is due to the utilization of the attention mechanism in the model to obtain better performance by enhancing the representational ability of nodes in heterogeneous networks.
- SpectralCF outperforms NeuMF on all three datasets, which indicates that the graph neural network-based recommendation model outperforms the general deep learning-

based model in terms of structure. Building the interaction between users and items as a bipartite graph can better explore the potential interests of users and enrich the embedding vectors of users and items.

- NGCF has better performance in comparison models. It defines the convolution operation directly on the spatial domain, without depending on the graph convolution theory. While improving flexibility, the performance of the spatial domain-based graph convolution recommendation model outperforms other methods.

- In comparison to NGCF, SGACF-EM improves the NDCG on the Yelp and Amazon datasets by 5% and 6%, respectively. SGACF utilizes a self-attention mechanism in the process of aggregating neighboring features to quantify the neighboring features of nodes according to their importance, thus improving the embedding representation of users and items. In order to make the calculation of attention coefficients more stable, the multi-headed self-attention mechanism is employed. The experimental results fully verify the rationality and effectiveness of the method proposed in this paper. CMN only utilizes the features of first-order neighbors, and SGACF mines higher-order collaborative signals based on higher-order connectivity, which illustrates the importance of higher-order connectivity principle in graph representation learning. SpectralCF and NGCF utilize spectral-based graph convolution and spatial domain-based graph convolution networks, respectively; both ignore the reliability of the feature propagation process between neighboring nodes and are prone to bring in noisy data. Moreover, in the training process of graph convolutional networks, the high degree of nodes tends to dominate the representation learning of the model. Therefore, SGACF joint self-supervised learning makes it easy for nodes with low degree to learn.

### 4.3.4. Analysis of The Impact of Comparative Learning

Contrastive learning is an implementation of self-supervised learning, which is mainly applied to solve the problem of little or no labeling of data. Therefore, its data-augmentation part is the main core. Although a great volume of data is generated inside recommendation domain, its data distribution shows a power–law distribution (i.e., long-tail problem). In the process of model training, nodes with high degree play a dominant role in representation learning, while nodes with low degree are very difficult to learn. Therefore, adding contrastive learning to the recommendation model to solve the long-tail problem is effective. Further, to make the model easy to train, the parameters are shared between graph neural networks for supervised and self-supervised tasks.

### 5. Ablation Analysis

To demonstrate the effectiveness of the adopted technique, this paper conducts ablation experiments on a multi-headed graph attention mechanism and a self-supervised task.

### 5.1. Impact of Multiple Attention Mechanism

This subsection analyzes the impact of the number of attention heads on performance in SGACF-EM. The use of a single-headed graph attention network overly focuses attention on its own position when encoding the current node features during the training process. To obtain more reliable attention coefficients, a multi-headed graph attention network is utilized to extract features from multiple perspectives. The experimental results are shown in Table 3. The model achieves better performance when the number of heads is 2. When the number of heads exceeds a certain number, the model effect is not improved, which is due to the limitation of the characteristics of the data itself.

**Table 3.** Effect of graph attention network with different number of heads on performance.

| Heads | Gowalla | | Yelp2018 | | Amazon | |
|-------|---------|------|----------|------|--------|------|
| | **Recall** | **Ndcg** | **Recall** | **Ndcg** | **Recall** | **Ndcg** |
| 1 | 0.2726 | 0.2411 | 0.2154 | 0.1397 | 0.1677 | 0.0914 |
| 2 | **0.2753** | **0.2423** | **0.2171** | **0.1402** | **0.1698** | **0.0925** |
| 4 | 0.2715 | 0.2405 | 0.2128 | 0.1374 | 0.1660 | 0.0903 |
| 8 | 0.2681 | 0.2344 | 0.2087 | 0.1316 | 0.1627 | 0.0842 |

The bold indicates the best result.

### 5.2. Optimization of Models by Contrastive Learning

To investigate the effect of self-supervised learning on model performance, this subsection performs an ablation analysis of SGACF. Table 4 displays the experiment's outcomes, where joint self-supervised learning in the recommendation model significantly improves the model's performance. This is attributed to the fact that self-supervised contrastive learning effectively mitigates the problem of sparse supervisory signals in the recommendation task by utilizing information about commonalities and differences between data pairs as supervised signals to auxiliary supervised learning through a data-augmentation strategy. Meanwhile, to explore the impact of self-supervised learning on the long-tail problem, in this paper, items are divided into 10 groups of different ranks based on the number of connected edges of individual nodes, and the total number of interactions in each group is the same. The larger the groupID, the higher the number of connected edges of individual nodes. This work compares the ability of SGACF-ED and SGACF-w/o to solve the long-tail problem, both of which have their network layers set to 3. Recall is the sum of each group. As shown in Figure 5, group 10 contributes a large proportion to recall, even though it contains a small number of items. It is thus clear that SGACF-w/o tends to recommend popular items, while long-tail items have fewer connected edges. It can be seen that the performance improvement of SGACF is to accurately recommend long-tail items.

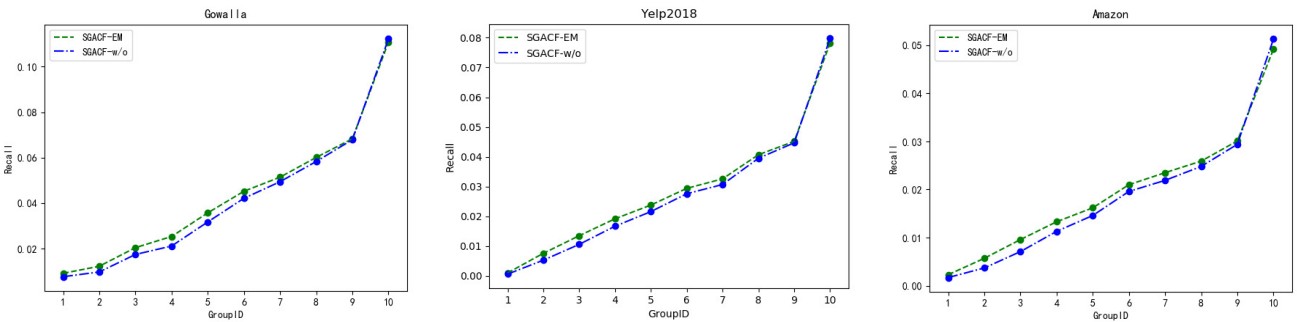

**Figure 5.** Performance comparison of different item groups SGACF-EM and SGACF-w/o.

**Table 4.** Impact of self-supervised learning on models.

| Method | Gowalla | | Yelp2018 | | Amazon | |
|--------|---------|------|----------|------|--------|------|
| | **Recall** | **Ndcg** | **Recall** | **Ndcg** | **Recall** | **Ndcg** |
| SGACF-w/o | 0.2680 | 0.2341 | 0.2037 | 0.1113 | 0.1622 | 0.0894 |
| SGACF-EM | **0.2753** | **0.2423** | **0.2171** | **0.1402** | **0.1698** | **0.0925** |

The bold indicates the best result.

### 6. Conclusions and Future Work

This paper proposes a self-supervised graph attention collaborative filtering for recommendation. It observes that graph convolution-based recommendation ignores the problem of importance between neighboring nodes, and thus employs graph attention networks to obtain reliable neighboring features. In this paper, self-supervised contrastive learning is utilized to auxiliary supervised learning, thus addressing the limitations of supervised

tasks in recommendation. The impact of three approaches to graph data augmentation on model performance is analyzed from the perspective of graph structure. Extensive experiments are conducted on three public datasets to demonstrate the advantages of the methods proposed in this paper in mitigating the long-tail problem and reducing the impact of interaction noise. To sum up, the contributions of this work are as follows: A multi-head graph attention network is utilized to aggregate neighbor features from multiple perspectives to reduce the impact of interaction noise on representation learning. Supervised tasks based on graph attention networks combined with self-supervised learning enhance representation learning via observing the properties of nodes in the interaction graph. Extensive experiments on three benchmark datasets demonstrate the effectiveness of the proposed self-supervised graph attention collaborative filtering for recommendation.

This work achieves better results, adopting the joint training of self-supervised contrast learning and supervised learning, but there are still limitations. The success of contrast learning-based representation learning relies heavily on well-designed data-augmentation strategies. In future research, we will focus on graph data-augmentation strategies in contrast learning. We hope to make contrast learning in recommendation more robust through data-augmentation strategies. For example, sampling-based data augmentation transforms both the adjacency matrix and the feature matrix.

**Author Contributions:** Methodology and writing—original draft preparation, J.Z.; review and validation, K.L., J.P. and J.Q. All authors have read and agreed to the published version of the manuscript.

**Funding:** This research was funded by the Natural Science Foundation of Hebei Province under grant No. F2022201009; the Hebei University High-level Scientific Research Foundation for the Introduction of Talent under grant No.521100221029; the Scientific Research Foundation of Hebei University for Distinguished Young Scholars under grant No. 521100221081 and the Scientific Research Foundation of Colleges and Universities in Hebei Province under grant No. QN2022107.

**Data Availability Statement:** All datasets are publicly available.

**Acknowledgments:** Thanks to the mentor for his careful guidance. Thanks to the anonymous reviewers for their insightful comments that improved the quality of this paper. Thanks to my girlfriend for encouraging me so much.

**Conflicts of Interest:** The authors declare no conflict of interest.

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
