# Peer review of "Self-Supervised Graph Attention Collaborative Filtering for Recommendation"

_electronics, doi:10.3390/electronics12040793_

Round 1

Reviewer 1 Report

Paper summary:

This paper introduces a self-supervised graph attention collaborative filtering for the recommendation. The self-supervised task mainly constructs supervised signals from the correlations within the input data and performs joint learning with the supervised task as an auxiliary task. Extensive experiments on three benchmark datasets demonstrate the effectiveness of the proposed method. 

Comments for author:

1. This paper is generally well-written. The logic is easy to follow.

2. The motivation is clear. There are still some limitations that need to be overcome, such as sparse supervised signals, long-tail problems, and interaction noise.

3. The ablation analysis shows that self-supervised learning can indeed improve performance, but there is no direct analysis that the addition of self-supervised learning mitigates the impact of the long-tail problem in recommendations (Lines 483 - 484). Could the authors design some experiments to verify this conclusion? In other words, if it's only because data augmentation mitigates the effect of high-degree nodes, can similar performance be achieved by applying data augmentation to supervised learning as well, without self-supervised learning?

4. The authors need to add citations for GBDT + LR model (Line 35) and Pairwise bayesian personalized ranking(BPR) loss (Line 274).

Some typos,

Line 24, an extra "combined"?

Figure 2. Self-Supervised Framework, but the title is Supervised Learning Framework.

Line 272, "the preference of user u for item i is:", missing of a formula?

Line 359, Please add the full name of the abbreviation: "NDCG"

Reviewer 2 Report

The paper proposes a self-supervised graph attention collaborative filtering for recommendation. 

The proposed method is described in details. Its effectiveness is illustrated by experiments on three public datasets.

The Introduction must be improved. I recommend authors describe the structure of their paper at the end of the Introduction section. It would be better if the authors state their contribution in the Conclusion instead of in the Introduction section.

I recommend the authors motivate why they introduce the three aspects (collaborative filtering-based recommendation, graph neural network-based recommendation, and self-supervised learning). 

To demonstrate the effectiveness of their model, the authors compare the proposed method with four methods. They do not motivate why they decided to compare their method with these four methods - NeuMF, CMN, SpectralCF, and NGCF.

The authors do not discuss the limitation of their study, their plans for future improvement of the model and other studies in the field.

Reviewer 3 Report

The authors propose a ”self-supervised graph attention collaborative filtering for recommendation“. But the primary objective function is still related to a supervised learning task, where self-supervised (contrasive) learning is incorporated as an auxiliary task to enhance the supervised learning task. Then further, authors compare methods on 3 datasets and claim they can resolve long-tail item recommendation problem by adding the contrastive loss.

1. How do the authors get the initial embedding for each dataset?

2. 2.1.3 has a missing equation at the end.

3. Can you give more evidence in section 3.3.4? Is it possible to theoretically prove it?

4. For contrastive learning part, normally people use the embedding for some downstream tasks, I am wondering what would happen in your case? First, do contrastive learning then fed to a classification layer.

Round 2

Reviewer 2 Report

The paper can be accepted